

# Metagenomic analysis of fungal assemblages at a regional scale in high-altitude temperate forest soils: alternative methods to determine diversity, composition and environmental drivers

Stephanie Hereira-Pacheco[1,2], Itzel Arias-Del Razo[2],
Alejandra Miranda-Carrazco[3,4], Luc Dendooven[4],
Arturo Estrada-Torres[2] and Yendi E. Navarro-Noya[1]

[1] Laboratorio de Interacciones Bióticas, Centro de Investigación en Ciencias Biológicas, Universidad Autónoma de Tlaxcala, San Felipe Ixtacuixtla, Tlaxcala, Mexico
[2] Estación Científica La Malinche, Centro Tlaxcala de Biología de la Conducta, Universidad Autónoma de Tlaxcala, Tlaxcala, Tlaxcala, Mexico
[3] Departamento de Ciencias Ambientales, Unidad Lerma, Universidad Autónoma Metropolitana, Lerma de Villada, Estado de México, Mexico
[4] Laboratorio de Ecología de Suelos, Departamento de Biotecnología y Bioingeniería, Cinvestav, Ciudad de México, Ciudad de México, Mexico

Corresponding author
Yendi E. Navarro-Noya,
yendiebenezer.navarro.n@uatx.mx

## ABSTRACT

**Background:** Understanding the diversity and distribution of fungal communities at a regional scale is important since fungi play a crucial role in ecosystem functioning. Our study used environmental metagenomics to determine fungal communities in mountainous forest soils in the central highlands of Mexico.

**Methods:** We used four different bioinformatic workflows to profile fungal assemblages, *i.e.*, Geneious+UNITE, single- and paired-end microbial community profiling (MiCoP), and Kraken2.

**Results:** The workflows yielded different results; one detected a higher abundance of ectomycorrhizal (EcM) and saprophytic fungi, while the other identified more saprophytic and pathogenic fungi. Environmental, vegetation, and geographical factors determined the spatial distribution of soil fungi at a regional scale. Potential hydrogen (pH), calcium (Ca), magnesium (Mg), and silt content were detected as common drivers of fungal communities across different datasets enriched towards a functional guild. Vegetation traits were found to be more influential in shaping symbiotrophic fungi composition than saprotrophic and pathogenic fungi. This highlights the importance of considering vegetation traits when studying fungal community diversity and distribution. Clustering patterns of sampling points near the volcanoes indicated shared environmental and vegetation characteristics. A weak but significant distance decay in taxonomic similarity revealed that dispersal limitation contributed to fungal community composition, although it was not the primary factor in this study. Overall, this study provides important insights into the challenges and opportunities of studying fungal communities at a regional scale using metagenomic data.

# INTRODUCTION

Soil fungi play a crucial role in providing various ecosystem services within forests. They contribute to nutrient recycling, enhance soil stability, facilitate soil carbon storage, improve soil water retention, and mediate biological interactions by controlling or producing plant and insect diseases. Additionally, fungi establish symbiotic associations with a wide range of organisms, including vascular plants, algae, cyanobacteria, and others. They also serve as habitats and food sources for numerous soil-dwelling organisms (*Bennett, 1998*; *Bhantana et al., 2021*; *Ward et al., 2022*).

Despite the importance of soil fungi, our understanding of their spatial patterns and the ecological factors shaping their communities remains context-specific, with limited generalizability across ecosystems and at larger scales. This requires further exploration, as a deeper understanding of these ecological factors could help preserve ecosystems in the face of environmental changes. Traditional methods, such as sampling fruiting bodies, have extensively studied diversity of fungal communities (*Peay & Bruns, 2014*). However, the use of environmental DNA (eDNA) analysis and high-throughput sequencing (HTS) in recent years has enabled the recovery of more comprehensive information about microbial diversity and composition (*Tedersoo et al., 2022*). This new approach shows great promise in enhancing our understanding of fungi in much greater detail.

Soil fungi exhibited different assembly patterns depending on the geographic scale. Dispersal limitation, land use, climate, seasonality, edaphic factors, and plant community were identified as the main drivers of the fungal distribution (*Glassman, Wang & Bruns, 2017*; *Lan et al., 2022*; *Li et al., 2022*; *Rasmussen et al., 2018*; *Tedersoo et al., 2020*; *Vasco-Palacios et al., 2020*; *Xing et al., 2021*; *Zhao et al., 2019*). Typically, climate, historical dispersal limitations, and the covariation of climate with vegetation and edaphic properties have a limited effect on the fungal distribution at a regional geographic scale, as these factors are intrinsic to larger geographic scales (*Lomolino, Riddle & Brown, 2006*; *Tedersoo et al., 2020*). However, different patterns might emerge in mountainous ecosystems, especially in highly biodiverse areas, such as the Trans-Mexican Volcanic Belt (TMVB) region. The TMVB is composed of the highest mountains (mostly volcanoes) in Mexico, and it is part of the Mexican Transition Zone. In this biogeographical zone, the neotropical and nearctic biotas overlap (*Arriaga-Jiménez, Rös & Halffter, 2018*). Additionally, it is a center of diversification for several species (*Moctezuma, Halffter & Escobar, 2016*; *Ruiz-Sanchez & Specht, 2013*). Specific environmental factors contribute to the high level of endemisms found in the TMVB, exceeding predictions based solely on general macroecological factors (*Rahbek et al., 2006*). The small surface area of these high mountain regions and their geographic isolation results in small and fragmented patches of vegetation (*Mastretta-Yanes et al., 2018*). Furthermore, the complex geological and volcanic history in the region made the edaphic characteristics very diverse (*Rodríguez et al., 2019*).

Soil fungal species and functional guilds respond differently to variations in soil characteristics, nutrient availability, and vegetation (*Peay, Baraloto & Fine, 2013*; *Bahram et al., 2020*; *Sui et al., 2022*). It is known that arbuscular mycorrhizal fungal (AMF) community composition is affected by abiotic variables (potential hydrogen (pH), rainfall, soil type, and land use), but not by geographical distance (*Hazard et al., 2013*; *Xu et al., 2017*). Both ectomycorrhizal (EcM) and saprotroph fungi are shaped by vegetation and forest type due to the presence of hosts, differences in litter composition or quality, and soil water-holding capacity (WHC) (*Peay, Baraloto & Fine, 2013*). Across different regions, the proportion of EcM plants had a strong effect on symbiotrophic fungi diversity in forests (*Hui et al., 2017*; *Kernaghan et al., 2003*; *Tedersoo et al., 2020*), and in some regions on saprotrophic fungi (*Khalid et al., 2024*). Additionally, the dispersal capacity can be different within functional groups of fungi. For example, EcM fungi have a high dispersal capacity due to their ability to disperse through wind-borne spores and animal vectors. In contrast, most AMF species are hypogeous, and while approximately one-third of the described species have the potential for aerial dispersal, some have large single-cell spores that limit long-distance dispersion (*Chaudhary et al., 2020*).

Although many studies focus on the patterns and drivers of the diversity of fungal groups, such as EcM (*Tedersoo, May & Smith, 2010*) and AMF (*Bouffaud et al., 2016*), only a few studies have investigated all fungi, and none have used a metagenomic approach on a regional scale. Metagenomic approaches have the advantage of not including a PCR step, which can introduce bias toward certain taxa. This study used HTS from eDNA and four bioinformatics workflows to determine fungal communities. The selected workflows were based on (1) the use of fungal-genome reference databases to map the reads, or (2) the mapping of reads against a genetic marker (barcode). Additionally, we selected methods that provide frequency or relative abundance data necessary for diversity analysis. The workflows yielded different results; one detected a higher abundance of ectomycorrhizal (EcM) and saprophytic fungi, while the other identified more saprophytic and pathogenic fungi. We used the results from the four workflows and determined how geographic distance, edaphic factors and nutrient content (environmental distance), and tree traits (vegetation) drove soil fungal community composition and diversity using datasets enriched towards functional guild at a regional scale in forest fragments dominated by two types of trees (deciduous and coniferous) in the east part of the TMVB in Mexico. We hypothesized that: (1) the spatial composition of EcM fungi was mainly affected by vegetation type and its characteristics while that of saprophytic fungi by the edaphic variables; and (2) the taxonomic composition of the database enriched with symbiotrophs would be best explained by tree composition and their traits, while when using the database enriched with saprotrophs, the taxonomic composition would be best explained by the edaphic factors and nutrient content.

Mountain forests in the neotropics are highly valuable ecosystems for their biodiversity and endemisms (*Salinas et al., 2021*). However, they are vulnerable to climate change, and many worldwide are under great pressure due to land use change, fragmentation, and disturbance of the natural vegetation (*Newbold et al., 2020*). More knowledge of the diversity and distribution of the species in this region is urgently needed to identify areas of

species richness (hotspots) and to designate protected areas to conserve their diversity. Additionally, this study explored various bioinformatic approaches for analyzing fungal communities using metagenomic data, which could aid in making informed decisions about bioinformatic workflows based on our findings.

# MATERIALS AND METHODS

## Study site and soil sampling

The study area was located in the states of Tlaxcala and Puebla (N 19°14′44.6821″ W 98°06′33.9509″–N 19°16′36.7785″ W 98°36′36.0215″). It included two National Parks, La Malinche (LMNP) and Iztaccíhuatl-Popocatépetl (IPNP), that are immersed in the most populated region of the continent (megalopolis), east of the TMVB. This region has a temperate subhumid climate. The average annual temperature is 14 °C, the average maximum temperature is around 25 °C and occurs in April and May, and the average minimum temperature is 1.5 °C in January. The average precipitation is 720 mm annually; with most rains in summer from June to September (*INEGI, 2024*). The elevation varies between 1,711 and 5,426 m above sea level (a.s.l.), and the original vegetation is temperate and subpolar coniferous forest, temperate and subpolar scrub, deciduous broadleaf forest, and temperate grasslands (*CONABIO, 2023*).

Sampling was done during the dry season of 2021. A total of 12 sampling sites distributed in six polygons (≈13 km$^2$), two sites per polygon were considered (Fig. 1, Table S1). Sampling areas were located within forest fragments dominated by *Abies*, *Pinus*, and *Quercus* genera species. At each sampling site, three transects of 50 m were defined, and a composite sample of soil was collected from each transect by taking 400 g of soil every 10 m along the entire length. The O horizon was removed, and the underlying 0–10 cm topsoil was sampled with a small hand spade. All samples from each transect were pooled and 5 mm sieved to create a single composite sample. As such, 36 soil samples (*N* = 36) were collected (6 polygons × 2 sites × 3 transects), and this field-based replication was maintained in the laboratory. Geographical coordinates were recorded at the beginning, middle, and endpoint of each transect. All soil samples were transported to the laboratory on ice and arrived within 60–90 min. Upon arrival, a sub-sample of 50 g was stored at −20 °C until extracted for DNA as described below, while the rest was air-dried and characterized.

## Soil characterization

The WHC was measured by adding three 20-g soil sub-samples separately to a plastic funnel with Whatman No. 42 filter paper, applying 50 ml water, and letting them drain overnight (*Cassel & Nielsen, 1986*). At the same time, another sub-sample of 20 g was used to determine the moisture content. The soil was weighed, and the water retained, including previously contained water, was considered the WHC. Soil pH and electrolytic conductivity (EC) were measured in 1:2 soil-H$_2$O suspension using a potentiometer/electrode (Hanna HI 2210; Hanna Instruments, Woonsocket, RI, USA) (*Thomas et al., 1996*). The nutrient concentrations were measured as follows: phosphorus (P) was measured in a photocolorimeter (Genesys$^{TM}$ 10S UV-VIS; Thermo Scientific, Waltham,
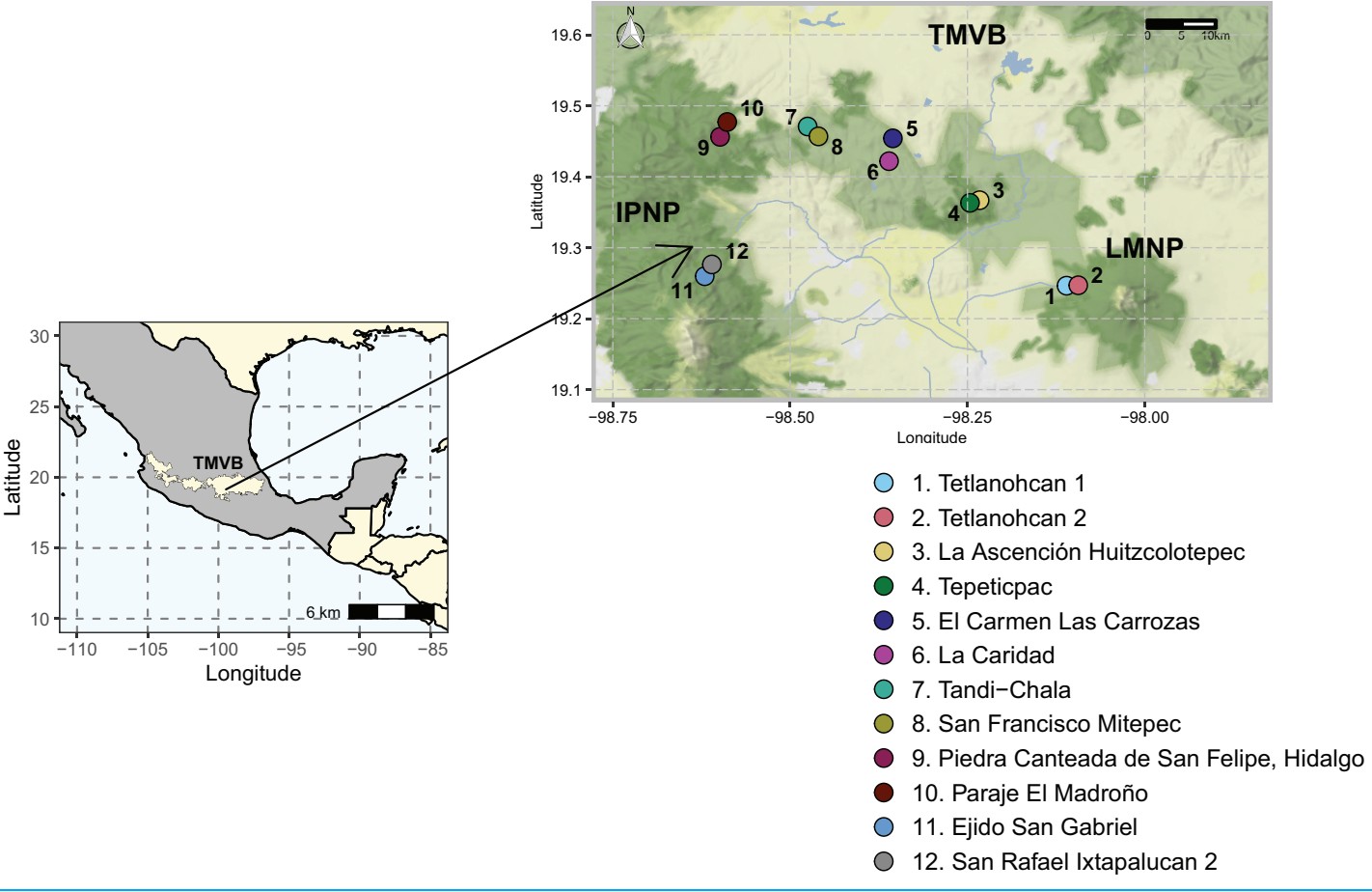

**Figure 1 Study area and 12 sampling sites between the National Parks *L. Malinche* (LMNP) and *Iztaccíhuatl-Popocatépetl* (IPNP) located on the east side of the Trans-Mexican Volcanic Belt (TMVB).** Map from Stadia Maps (stadiamaps.com).

MA, USA), potassium (K) in a flame emission spectrophotometer (Cole-Parmer 2655-00 digital flame analyzer; Cole-Parmer Instrument Co., IL, USA), magnesium (Mg), calcium (Ca), iron (Fe), copper (Cu) and manganese (Mn) with an atomic absorption spectrometer (model SAVANTAA; GBC Scientific, Melbourne, Australia). Total nitrogen (N) was measured by the Kjeldahl method (*Bremner, 1996*), and the soil particle size distribution by the hydrometer method as described by *Gee & Bauder (1986)*. Soil organic matter (SOM) was determined using the chromic acid titration method described by *Walkley & Black (1934)*.

## Characterization of the vegetation

The vegetation was characterized at each sampling site. A 50 × 25 m quadrant was delineated within each transect, and individual trees were identified at the species level. Total height was measured using a forest clinometer. Coverage was determined using the crown diameter, and basal area was computed based on the diameter at breast height for each tree. Proportions were obtained by genus and type of tree (*i.e.*, conifer or

broadleaf). Means of measured characteristics were calculated by sampling sites, tree genus, and tree type.

## Soil DNA extraction and high-throughput sequencing

Metagenomic DNA was extracted using the DNeasy PowerSoil Pro Kit (QIAGEN, Hilden, Germany) according to the manufacturer's instructions. Following the manufacturer's instructions, library construction was done with the Kapa Biosystems (Roche, CA, USA) kit. The size-selected libraries were amplified with three cycles of PCR and run on a Fragment Analyzer (AATI, Ankeny, IA, USA) to confirm the absence of free primers and adaptor dimers, and the presence of DNA of the expected size range. Libraries were pooled in equimolar concentration and quantitated by qPCR on a Bio-Rad CFX Connect Real-Time System (Bio-Rad Laboratories Inc., Hercules, CA, USA). Sequencing was done in an Illumina Novaseq 6000 platform, using a 150 bp × 2 run, resulting in ~30 M reads per sample. Library preparation and sequencing were done by the "*Centre d'expertise et de services Génome Québec*" (Quebec, Canada).

## Sequence processing and taxonomic analysis

First, all reads were filtered to remove human sequences using BBMAP v.38.18 (*Bushnell, 2014*). Then, a quality score filter was applied using TRIMMOMATIC v.0.39 to remove low-quality reads and/or those containing Illumina adaptors using the following parameters: ILLUMINACLIP:2:30:10; SLIDINGWINDOW:4:20; and MINLEN:30 (*Bolger, Lohse & Usadel, 2014*). The resulting high-quality sequences were analyzed using various bioinformatic workflows to determine fungal taxonomic composition. Three read-based pipelines were employed: Kraken2, MiCoP (Microbial Community Profiling), and Geneious+UNITE (Fig. 2). The Kraken2 workflow (*Wood, Lu & Langmead, 2019*) used a k-mer based approach and a machine learning classification algorithm against the PlusPFP index database (https://benlangmead.github.io/aws-indexes/k2) that contains fungi taxa constructed within the Kraken2 software. The abundance of taxonomic groups was determined with Bracken v.2.5 (*Lu et al., 2017*). Within the MiCoP workflow (repository cloned in November 2021; *LaPierre et al., 2019*) two options were used: single-end and paired-end. First, the reads whether single or paired, were mapped to a reference database of complete fungal genomes using the BWA-MEM v.0.6 software (*Li, 2013*). Subsequently, a two-stage process was used to classify the reads. Initially, all uniquely mapped reads were classified using MiCoP's built-in database and the abundance of each genome was determined in each sample based on these reads. Multi-mapped reads were assigned probabilistically to one of the genomes with probabilities proportional to the abundance obtained from uniquely-mapped reads of those genomes (*LaPierre et al., 2019*). The Geneious+UNITE workflow (*Bolyen et al., 2019*) was followed as suggested by *Smith et al. (2020)*. First, reads were mapped against a modified UNITE (dynamic version 9, http://unite.ut.ee/) reference database (*Nilsson et al., 2019*) using the Geneious read mapper with 'Medium-Low Sensitivity/Fast sensitivity' settings and iterated twice (*Kearse et al., 2012*). The mapped reads were imported into the QIIME2 v.2022.2, dereplicated using q2-vsearch (*Rognes et al., 2016*), and filtered with 'qiime quality-control exclude-seqs' against the
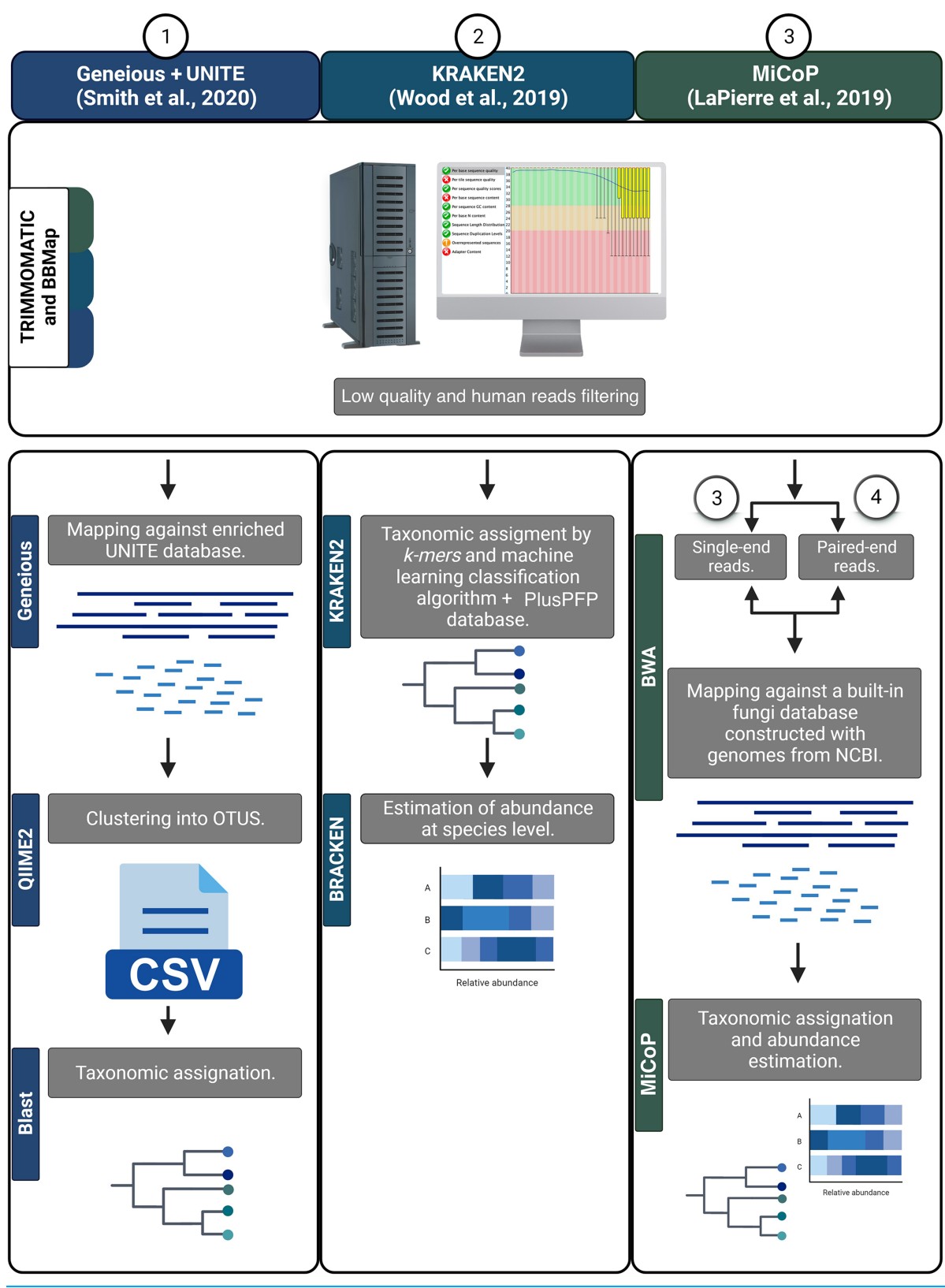

**Figure 2** Bioinformatic workflows to determine fungal taxonomic profiles.

UNITE database. The remaining sequences were clustered *de novo* into OTUs at a 97% similarity using q2-vsearch (*Rognes et al., 2016*; *Smith et al., 2020*). Taxonomic assignment was done with BLAST at 98% of identity, and unassigned reads were filtered out.

As the taxonomic unit obtained by the various pipelines was either taxonomic species or OTUs, we will refer to them as 'features'. Tables displaying the frequency counts generated from each workflow will be referred to as 'feature-tables'. To infer and annotate the ecological guilds of the feature-tables derived from the four workflows, we used the FUNGuild v.1.0 software and enriched the annotation manually (*Nguyen et al., 2016*).

## Statistical analysis

All statistical analyses were done using R v.4.3.1 (*R Core Team, 2023*), and figure generation with packages ggplot2 (v.3.4.4) and ggpubr (v.0.6.0) (*Wickham, 2011*; *Kassambara, 2020*). To create maps, we used the ggmap package within the Stadia Maps using a personal API key (*Kahle & Wickham, 2013*). Heatmaps representing the relative abundance of taxonomic annotations were done with the ComplexHeatmap package v.2.14.0 (*Gu, Eils & Schlesner, 2016*). All figures were created using colorblind-friendly palettes from the R package rcartocolor v.2.1.1 (*Nowosad, 2023*).

Alpha diversity indexes were calculated using Hill numbers. This approach provides intuitive measures in the "effective number of features", which facilitate interpretation and comparison. Hill numbers modulate the weight of species abundance with the parameter q (diversity order), *i.e.*, q = 0 corresponds to the total number of features (richness), q = 1 corresponds to frequent features (equivalent to Shannon exponential entropy), and q = 2 to dominant features (equivalent to the reciprocal of the Simpson measure) (*Chao, Chiu & Jost, 2014*). The Hill numbers were calculated using the hillR package v.0.5.1 (*Li, 2018*). Kruskal-Wallis tests were performed to examine differences between sites with the 'kruskal.test' function within R (stats default package).

To compare taxonomic compositions within the Hill numbers framework, dissimilarity community matrices were computed from each workflow's feature-tables at order q = 1 (accounting for frequent features and equivalent to Classic Horn dissimilarity index; *Chao, Chiu & Jost, 2014*) using the function 'hill_taxa_parti_pairwise' of the hillR package. Bray-Curtis dissimilarity matrices were calculated within phyloseq as this metric is widely used in similar ecological studies (*McMurdie & Holmes, 2013*). The similarity was calculated when necessary by subtracting the dissimilarity value from one (1-dissimilarity). A permutational multivariate analysis of variance (perMANOVA) was done with the adonis2 function (*n* = 999) to test community differences between sites (*n* = 12), and a permutational multivariate homogeneity of group dispersion analysis (PERMDISP) with the betadisper function. Both analyses were done with the vegan package v.2.5.3 (*Oksanen et al., 2017*). Additionally, a principal coordinate analysis (PCoA) was plotted to explore dissimilarity patterns of fungal communities obtained with each pipeline with the 'pcoa' function from ape v.5.5 (*Paradis, Claude & Strimmer, 2004*).

To determine the effect of the geographic distance on fungal communities, a geographical distance matrix was obtained using geographical coordinates with the function 'distm' from the geosphere package v.1.5-14 (*Hijmans, 2021*) (Table S1).

**Table 1 Summary of the frequencies obtained with different bioinformatic workflows.**

| Bioinformatic workflow | Total frequencies | Number of features[a] | Minimum frequency by sample | Maximum frequency by sample | Mean frequency by sample | Median frequency by sample | Mean Good's coverage (%) |
|---|---|---|---|---|---|---|---|
| Geneious+ UNITE | 34,739 | 3,138 | 395 | 2,059 | 965 | 848 | 79.83 |
| Single MiCoP | 93,589 | 210 | 1,194 | 5,043 | 2,600 | 2,392 | 99.32 |
| Paired MiCoP | 1,281,436 | 234 | 21,654 | 52,713 | 35,595 | 35,008 | 99.94 |
| Kraken2 | 1,021,469 | 87 | 17,680 | 42,316 | 28,374 | 27,185 | 100 |

**Note:**
[a] Taxonomic species (single and paired MiCoP, and Kraken2) or operational taxonomic unit at 97% of similarity (Geneious+UNITE).

Dispersal limitation was measured as an increasing dissimilarity in composition with geographic distance, *i.e.*, distance decay (*Kivlin et al., 2014*). Mantel tests with the Spearman method were used to evaluate the association between community similarity and geographic distance using the 'cor.test' function and linear regressions with the 'lm' function (stats default R package). Partial Mantel tests were applied to assess correlations between environmental and vegetation matrices (Euclidean distances) and community matrices conditioned on spatial distances.

The following steps were taken to determine the effect of environmental factors (soil characteristics and nutrient content) and vegetation on fungal taxonomic compositions. First, variables were standardized (proportion data were log normalized, centered, and scaled). Next, collinearity among the variables was assessed using correlation plots generated with the corrplot package v.0.92. Variables exhibiting high correlation (Pearson correlation: $|r| > 0.7$) were subsequently removed from the analysis (*Gao et al., 2020*; *Wei & Simko, 2021*). To select the variables explaining most of the variance of fungal communities: (1) feature-tables were Hellinger-transformed, (2) a constrained correspondence analysis (CCA) was done with the transformed data followed by a fit of environmental factors onto an ordination with the function 'envfit', and (3) a redundancy analysis (RDA) was done with the transformed data followed by a forward selection using 'capscale' function with the Horn and Bray-Curtis distances that account for abundance. Both CCA and RDA were done with 999 permutations and variables retained at $p < 0.05$. All ecological analyses were done with the vegan package.

Raw sequences are available at the NCBI Sequence Read Archive under BioProject PRJNA875082. All scripts and metadata for statistical analyses and visualizations are available at https://zenodo.org/doi/10.5281/zenodo.13629859.

## RESULTS

### Alpha diversity patterns of the fungal communities obtained by different workflows in the soils across sites

After implementing the described workflows, the total number of reads varied markedly (Table 1). The strategy that recovered the most reads and/or total frequency was Paired MiCoP, with 1,281,436 reads, and Geneious+UNITE, the least, with 34,739 counts. The

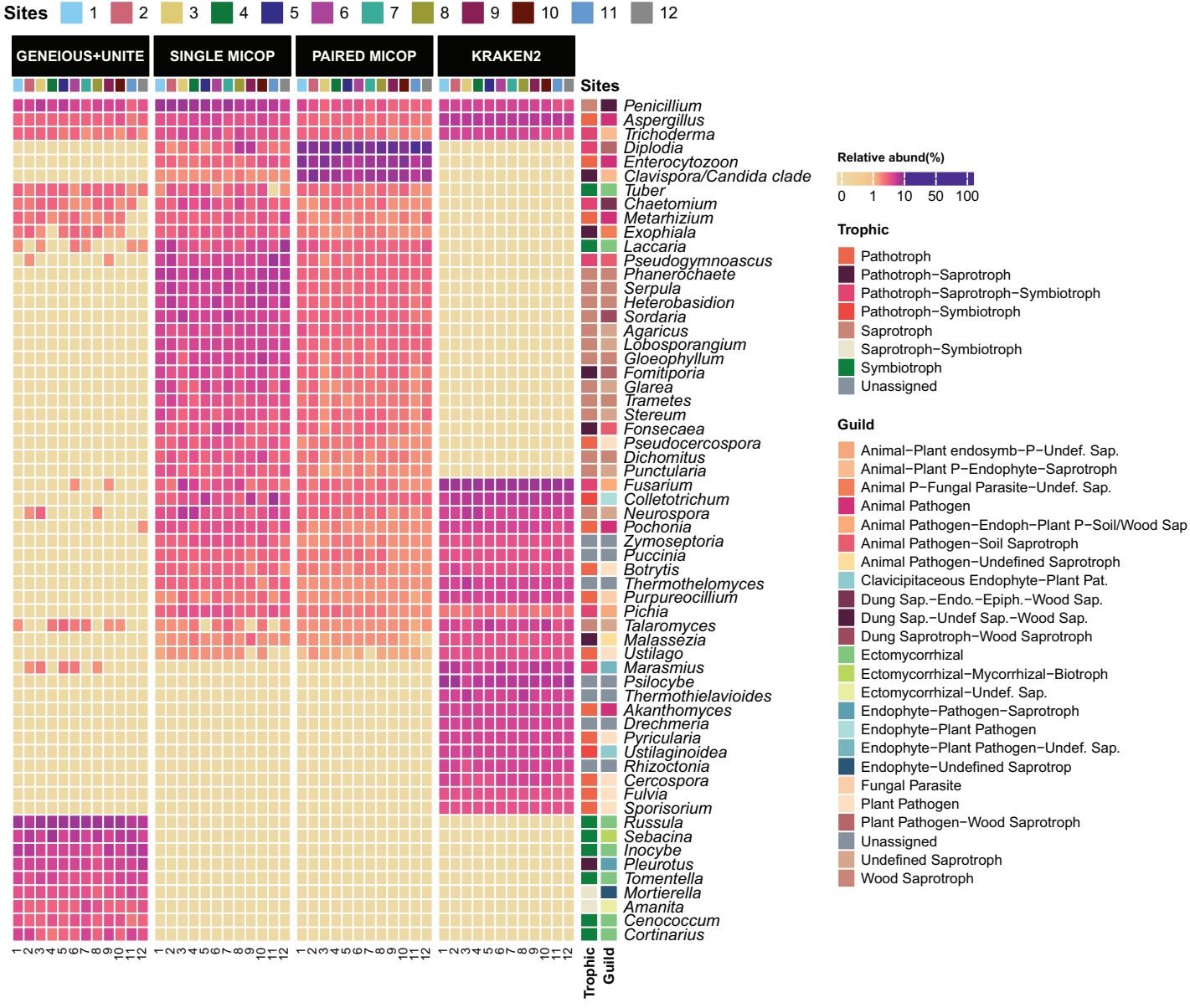

**Figure 3 Heatmap of the relative abundance of the 50 most abundant genera across the sampling sites (1–12) obtained by different bioinformatic workflows,** *i.e.*, **GENEIOUS+UNITE, SINGLE and PAIRED MICOP and KRAKEN2 (see Fig. 2).** Annotations on the right indicate the guild and trophic assignment with FUNGuild. Colored numbers refer to the different sampling sites and their location is given in Fig. 1.

number of features detected varied across the different workflows. Geneious+UNITE identified the highest number of features (3,138) and Kraken2 the least, with 87. The mean Good's coverage by workflow ranged from 79% to 100% of the total diversity.

The different workflows gave contrasting results that affected the calculation of the alpha diversity. The workflows, except Kraken2, which had a low variation, coincided in sites 11 and 12 (coniferous forest located in the IPNP), which had the lowest richness ($q = 0$) (Fig. S1). Diversity values, Hill numbers at $q = 1$ and 2 (frequent and dominant

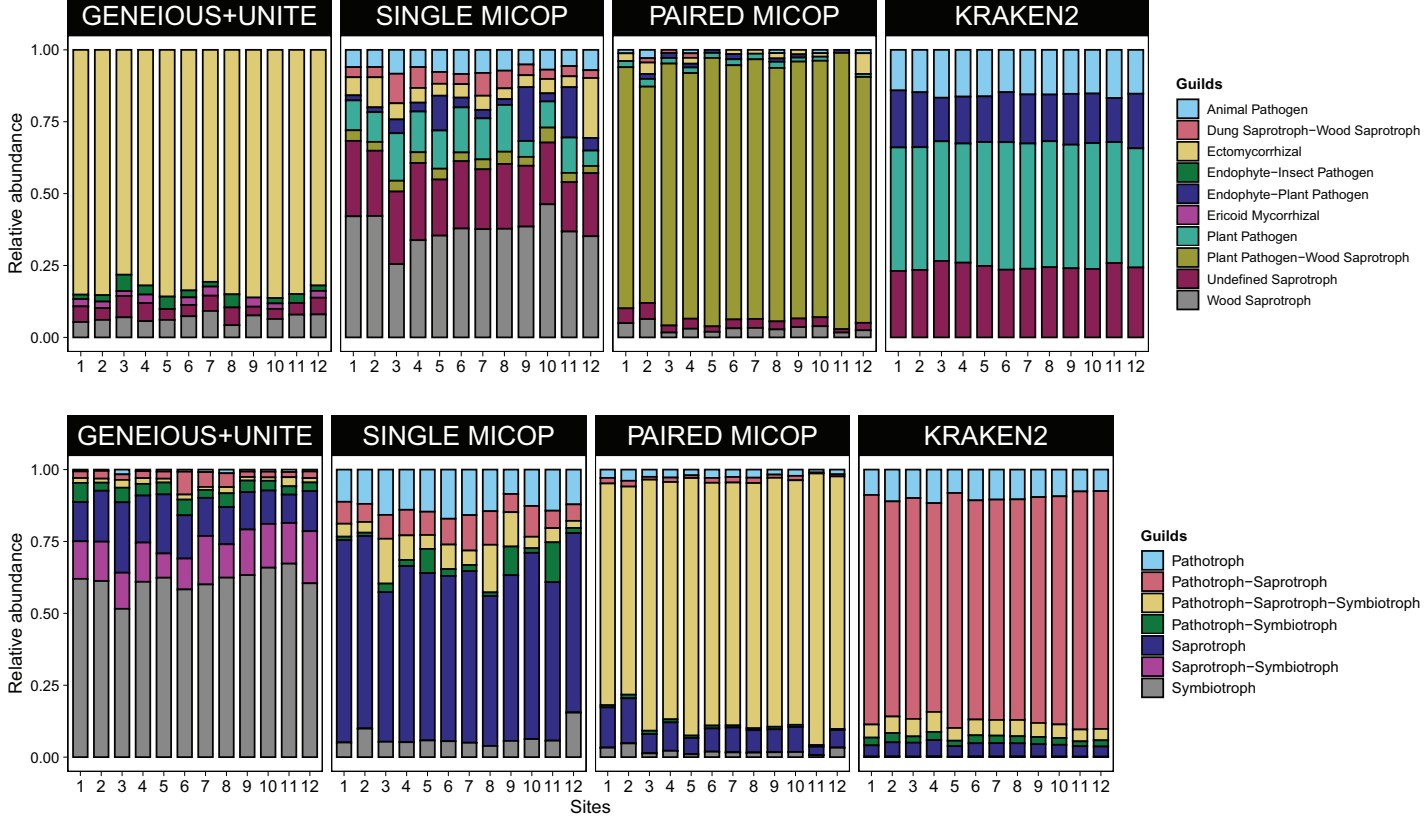

**Figure 4 Bar plots of the relative abundance of the 20 most abundant in (A) guilds and (B) trophic groups across different sites (1–12) and workflows, *i.e.*, GENEIOUS+UNITE, SINGLE and PAIRED MICOP and KRAKEN2 (see Fig. 2).** Numbers in the x-axis refer to the different sampling sites and their location is given in Fig. 1.               

features), were similar across sites when using the Geneious+UNITE, Paired MiCoP, and Kraken2 methods, while Single MiCoP displayed distinct patterns across the sites.

## Taxonomic and functional assignment of fungi by sites affected by workflow

The taxonomic composition and abundance varied considerably with the workflow (Fig. 3). The most abundant genera differed across the different approaches, with only the genera *Aspergillus, Penicillium*, and *Trichoderma* detected by all workflows. For instance, Geneious+UNITE workflow detected *Russula* (mean relative abundance 15.0 ± 8.3%) and *Sebacina* (5.7 ± 4.1%) as the most abundant genera, while the Single MiCoP *Penicillium* (10.1 ± 4.7%) and *Phanerochaete* (5.7 ± 2.0%) (Fig. 3). Paired MiCoP identified *Diplodia* as the dominant genus with a mean relative abundance of 57.3 ± 18.1%, followed by *Enterocytozoon* (16.9 ± 11.8%), while Kraken2 identified *Fusarium* (14 ± 3.9) as the most dominant genus followed by *Psilocybe* (7.6 ± 2.9).

Consistent with the analyses of taxonomic composition, functional assignments to guilds and trophic groups depended largely on the workflow used. The Geneious+UNITE workflow identified a greater number of taxonomic groups assigned to symbiotrophs, including ectomycorrhizal fungi genera, such as *Russula, Sebacina, Inocybe*, and

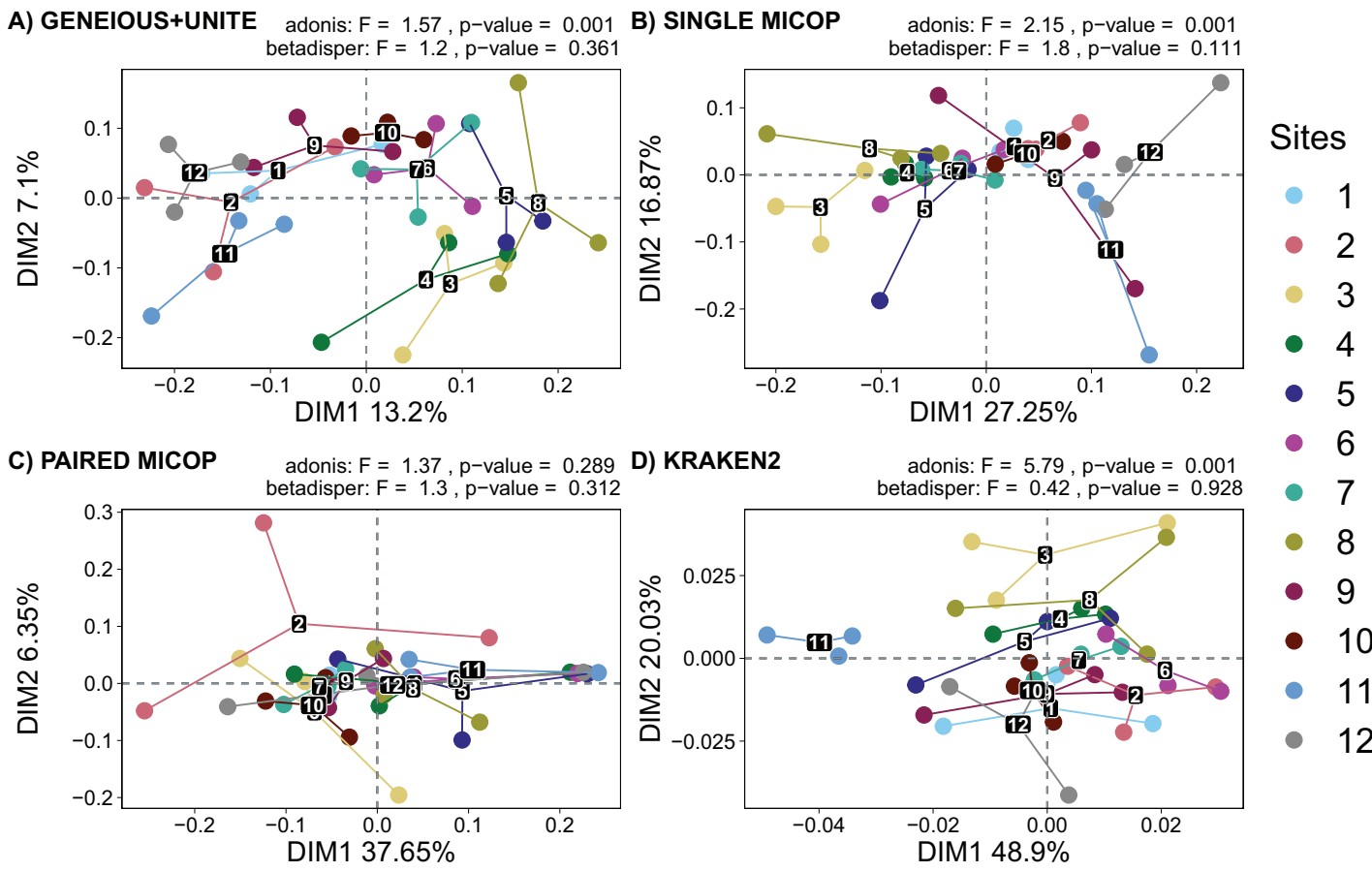

**Figure 5 Unconstrained principal coordinate analysis (PCoA) with Horn dissimilarities of the taxonomic structure of fungal communities (operational taxonomic units at 97% similarity or species) using different workflows: (A) GENEIOUS+UNITE, (B) SINGLE MICOP, (C) PAIRED MICOP and (D) KRAKEN2 (see Fig. 2).** Variations in variance and homogeneity among sampling sites were determined with a permutational analysis of variance (adonis) and a permutational multivariate homogeneity of group dispersions analysis (betadisper). Numbers refer to the different sampling sites and their location is given in Fig. 1.               

*Tomentella*, as well as some saprotrophs such as *Pleurotus* (Figs. 3 and 4). The Paired MiCoP approach exhibited a higher abundance of groups assigned to pathogens (*e.g.*, *Clavispora/Candida* clade, *Enterocitozoon*, and *Diplodia*). Single MiCoP detected a higher abundance of saprotrophs, especially wood saprotrophs (*e.g.*, *Phanerochaete* and *Serpula*), while symbiotrophs and pathogens had a lower abundance. Kraken2 identified saprotrophs and pathotrophs, particularly plant pathogens, such as the pathogen/saprotroph *Fusarium* (Figs. 3 and 4).

## Dissimilarity patterns of soil fungal communities across sampling sites

Horn and Bray-Curtis metrics showed similar patterns (Figs. 5 and S2). The sampling sites clustered into three groups, *i.e.*, sites 1, 2, 9, 11, and 12, which are closer to the volcanoes of the National Parks *L. Malinche* and *Iztaccíhuatl-Popocatépetl*, sites 7 and 10 as a transition cluster, and sites 3, 4, 5, 6 and 8 which are in the transects between the National Parks.

These groups were more defined with Geneious+UNITE and Single MiCoP workflows and less with Kraken2 and Paired MiCoP workflows.

The analysis of dispersion showed that the variance was homogeneous across the sampling sites using both dissimilarity metrics ($p > 0.05$). The perMANOVA test showed that the sampling site had a significant effect on the fungal taxonomic composition obtained by all workflows ($p < 0.05$), except for Paired MiCoP with the Horn dissimilarity index (Fig. 5C).

## Fungal communities' similarity as affected by geographical distance

Weak but significant ($p < 0.05$) negative correlations were observed between Horn similarities of fungal communities obtained by the different workflows and the spatial distance (Fig. 6). Absolute values of the Spearman Mantel correlations were in the order Single MiCoP ($r = |-0.241|$) > Geneious+UNITE ($r = |-0.228|$) > Kraken2 ($r = |-0.193|$) > Paired MiCoP ($r = |-0.0989|$). A different pattern was found with the Bray-Curtis similarities, although the negative correlations and significance were maintained: Geneious+UNITE ($r = |-0.260|$) > Single MiCoP ($r = |-0.213|$) > Kraken2 ($r = |-0.212|$) > Paired MiCoP ($r = |-0.177|$) (Fig. S3).

## Fungal communities' similarity as affected by environmental and vegetation characteristics

An exploratory PCA of environmental characteristics, *i.e.*, soil properties and nutrient content, showed two main clusters (Fig. S4). The first cluster comprised site samplings 1–2 and 9–12, located closer to the volcanoes in the National Parks *La Malinche* and *Iztaccíhuatl-Popocatépetl*. The second cluster consisted of sites 3–8, part of the transect between the National Parks. The first cluster was more spread out when compared to the second one, suggesting greater variations in environmental characteristics within the mountainous regions. Also, the first cluster was defined by higher P, N, and SOM content, and the second by higher WHC, pH, and content of clay, silt, Fe, K, and Cu.

Sites 1 and 12 were dominated by coniferous trees, sites 3–7 and 10–11 by broadleaf trees, while sites 2, 8–9 by conifers and broadleaves (Fig. S5A). Considering the tree genera, sites 1, 3–5 and 10–11 were dominated by one genus, the others by two or three (Fig. S5B). Accordingly, when considering vegetation and its characteristics, sampling sites were clustered into two groups, *i.e.*, sites 1–2 and 9–12 in one cluster and sites 3–8 in another (Fig. S6).

Correlations between the selected environmental and vegetation variables considered are given in Fig. S7. The CCA showed that the environmental variables that significantly ($p < 0.05$) shaped the fungal communities differed across the bioinformatic workflow. Concentrations of Ca, loam, Mg, K, P and SOM, and pH and WHC significantly affected the soil fungal composition obtained by Geneious+UNITE workflow (Table 2). These variables, except the content of K, and P, shape the soil fungal composition obtained by Single MiCoP. For Paired MiCoP, the significant variables were SOM, Mg, P and N content, and pH, while for Kraken2 SOM, Ca and loam content, and pH. When accounting

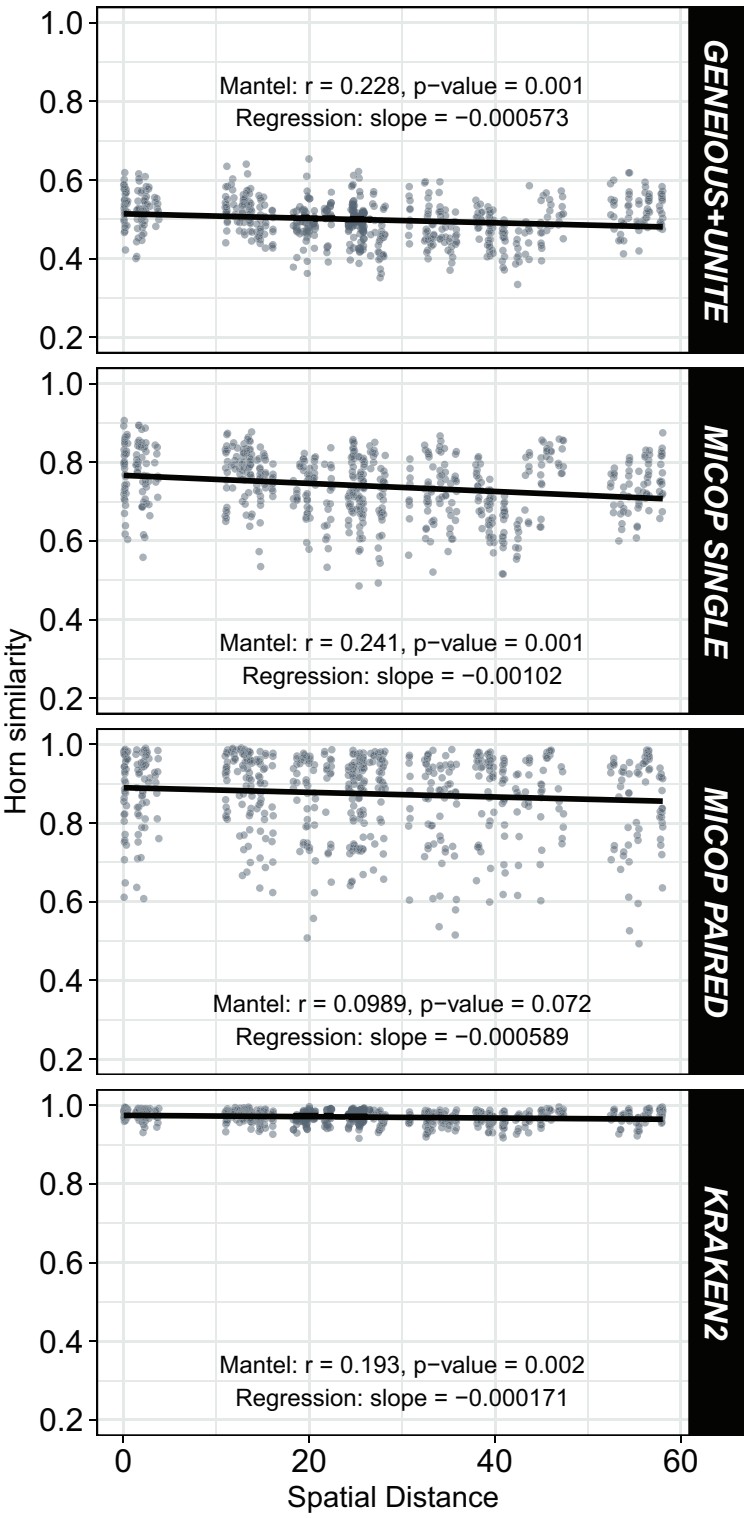

**Figure 6 Relationship between Horn similarity of fungal communities obtained by different bioinformatic workflows (GENEIOUS+UNITE, SINGLE and PAIRED MICOP and KRAKEN2; see Fig. 2) and geographical distance.** Mantel tests with Spearman's correlation were used to determine the association between community similarity and geographic matrices.

**Table 2** *p*-values of environmental variables resulted from 'envfit' using a canonical correlation analysis (CCA) on the Hellinger-transformed sequence data.

| Environmental variable | Geneious+UNITE | Single MiCoP | Paired MiCoP | Kraken2 |
|---|---|---|---|---|
| OM[a] | **0.015**[b] | **0.004** | **0.029** | **0.001** |
| Ca | **0.001** | **0.001** | 0.492 | **0.001** |
| Silt | **0.001** | **0.001** | 0.060 | **0.033** |
| WHC[c] | **0.001** | **0.001** | 0.105 | 0.378 |
| Fe | 0.064 | 0.481 | 0.118 | 0.451 |
| K | **0.038** | 0.080 | 0.786 | 0.786 |
| Mg | **0.001** | **0.001** | **0.015** | 0.084 |
| pH | **0.001** | **0.001** | **0.001** | **0.001** |
| Cu | 0.056 | 0.322 | 0.256 | 0.383 |
| Clay | 0.296 | 0.335 | 0.621 | 0.312 |
| P | **0.021** | 0.114 | **0.051** | 0.577 |
| N | 0.240 | **0.001** | **0.012** | 0.152 |

**Notes:**
[a] OM, Organic material.
[b] Values in bold are significant ($p < 0.05$) and were retained for environmental distances analysis.
[c] WHC, water holding capacity.

for the abundance of fungal groups (Horn, Table S2, and Bray-Curtis, Table S3), Ca and SOM content, and WHC and pH, were significant when using Geneious+UNITE. For Single MiCoP the significant variables were WHC, pH and N content with the Horn distance matrix, and WHC, pH, and Mg and Ca content with Bray-Curtis.

The CCA plots with soil fungal groups as determined with the Geneious+UNITE and Single MiCoP workflow and the environmental variables gave more clearly separated clusters than those determined with Paired MiCoP and Kraken2 workflows (Fig. 7). These patterns were consistent, but less defined for Paired MiCoP and Kraken2 when accounting for abundance (Horn and Bray-Curtis distances) (Figs. S8 and S9). The clustering pattern observed in this analysis was consistent with previous findings. Specifically, the sites located near the National Parks (1–2 and 9–12) formed one group, while the sites in the middle of the transect (3–8) constituted another group.

The significant vegetation traits also varied across workflows. The Geneious+UNITE workflow gave the highest number of significant traits with 11 in the CCA while Kraken2 had only 4 (Table 3). However, other results were obtained when accounting for abundance using Horn and Bray-Curtis distances (Tables S4 and S5). Similar clustering patterns for the different sites were obtained across workflows for vegetation traits (Figs. 8, S10 and S11).

Partial Mantel tests exhibited higher and positive correlation values between fungal communities and the environmental distance matrix (ranging from 0.112 to 0.260 among the different workflows using Horn dissimilarity and from 0.164 to 0.280 using Bray-Curtis dissimilarity; Tables S6 and S7) than with the vegetation distance matrix (ranging from −0.086 to 0.140 using Horn dissimilarity and from −0.098 to 0.178 using Bray-Curtis dissimilarity; Tables S6 and S7).

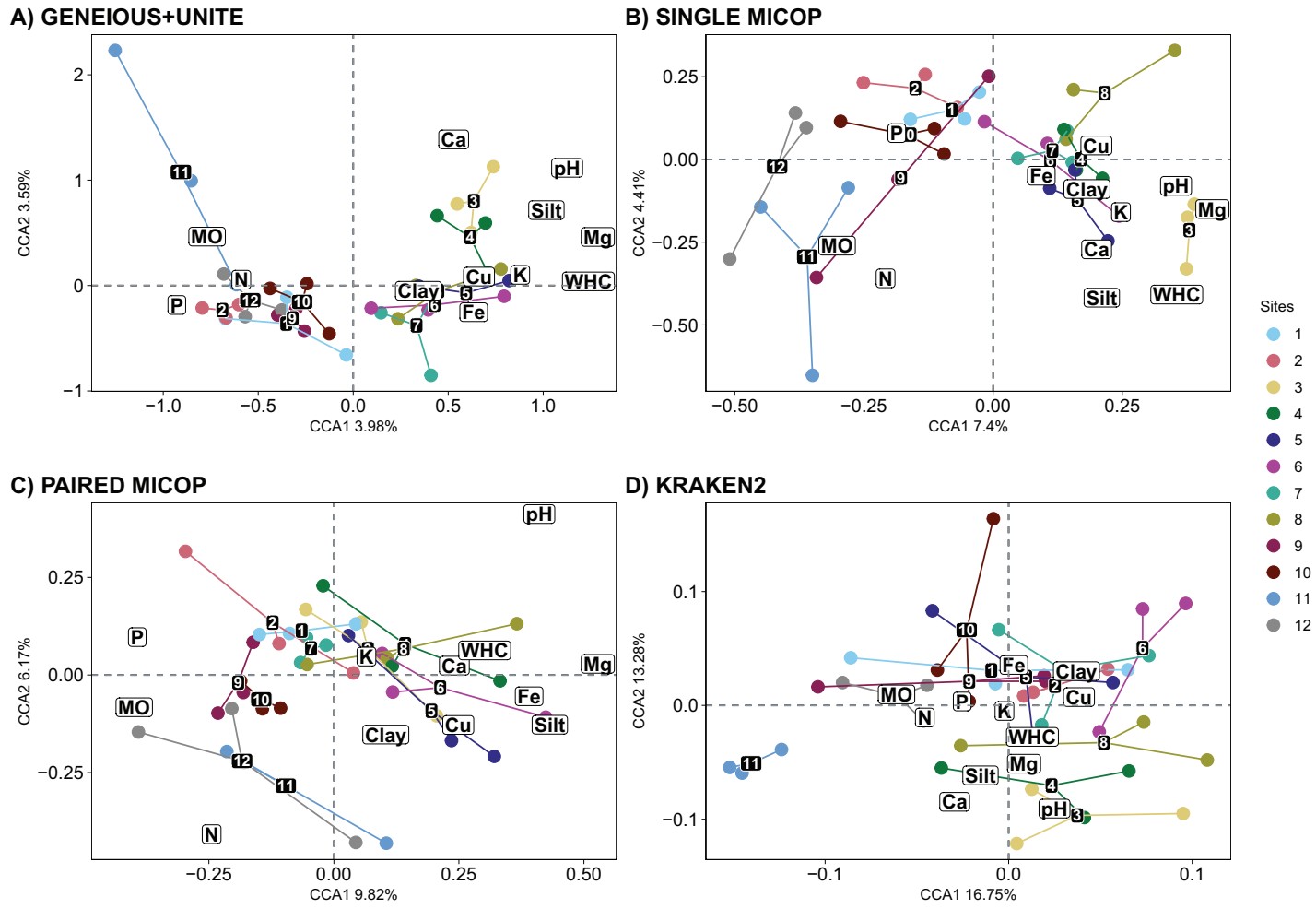

**Figure 7 Canonical correspondence analysis (CCA) with the composition of soil fungal groups and environmental characteristics,** *i.e.*, **physicochemical soil properties and nutrient content, using different workflows: (A) GENEIOUS+UNITE, (B) SINGLE MICOP, (C) PAIRED MICOP and (D) KRAKEN2 (see Fig. 2).** Numbers refer to the different sampling sites and their location is given in Fig. 1.

## DISCUSSION

In this study, we analyzed fungal communities in neotropical mountainous forest soils using environmental metagenomics. We aimed to identify the environmental, vegetation, and geographical factors controlling the spatial distribution of soil fungi on a regional scale. We used four different bioinformatic workflows for the fungal taxonomic profiling, and the results varied depending on the workflow used. This highlights the importance of a meticulous workflow selection and the use of suitable tools for an accurate analysis of fungal communities. It is important, however, to note that the fungal groups detected by all the workflows were characteristic of forest soils. Therefore, while misidentifications might be possible, the results were biased toward specific taxonomic or functional groups. For instance, when using the Geneious+UNITE workflow, we observed a high abundance of symbiotrophic fungi, such as the genera *Russula, Sebacina, Inocybe, Tomentella*, and *Cortinarius*. These are dominant and/or highly diverse EcM fungi in coniferous and

**Table 3 *p*-values of vegetation traits resulted from 'envfit' from canonical correlation analysis (CCA) on the Hellinger-transformed sequence data.**

| Vegetation variable | Geneious+ UNITE | Single MiCoP | Paired MiCoP | Kraken2 |
|---|---|---|---|---|
| Proportion of broadleaf | **0.009**[a] | **0.006** | 0.090 | 0.878 |
| Proportion of *Abies* | **0.001** | **0.001** | **0.001** | **0.018** |
| Proportion of *Pinus* | **0.001** | **0.001** | 0.376 | 0.629 |
| Proportion of *Salix* | **0.005** | 0.054 | **0.041** | **0.032** |
| Proportion of *Arbutus* | **0.001** | **0.001** | **0.012** | 0.082 |
| Proportion of *Alnus* | **0.001** | 0.075 | **0.014** | 0.088 |
| Proportion of *Juniperus* | **0.039** | **0.009** | 0.052 | **0.009** |
| Proportion of *Prunus* | 0.584 | 0.126 | 0.482 | 0.941 |
| Mean of total height | **0.003** | **0.001** | 0.102 | 0.061 |
| Mean of total coverage | 0.130 | 0.118 | 0.179 | 0.168 |
| Coverage of *Pinus* | **0.007** | **0.025** | **0.032** | 0.361 |
| Coverage of *Arbutus* | **0.018** | **0.035** | 0.052 | 0.072 |
| Coverage of *Quercus* | **0.002** | **0.001** | **0.007** | 0.079 |
| Coverage of *Juniperus* | 0.217 | 0.133 | 0.169 | **0.029** |

**Note:**
[a] Values in bold are significant ($p < 0.05$) and were retained for environmental distances analysis.

broadleaf forests from the TMVB (*Reverchon et al., 2012*) and from other Nearctic (*Barroetaveña, Cázares & Rajchenberg, 2007*; *Kennedy et al., 2012*) and Neotropical areas (*Carranza et al., 2018*). They play a crucial role in forest ecosystems by forming mutualistic associations with tree roots, facilitating nutrient exchange, and enhancing plant growth and survival (*Behie & Bidochka, 2014*). The Single MiCoP approach revealed a higher abundance of saprotrophic fungi, particularly *Serpula*, commonly found on dead wood of various coniferous tree species (*Carlsen et al., 2011*), and a lower abundance of EcM fungi, such as the genus *Tuber*, which has been reported to aid in host nutrient acquisition (*Guevara et al., 2013*). In contrast, the Paired MiCoP and Kraken2 workflows detected pathogens including *Diplodia*, *Clavispora/Candida* clade, *Fusarium*, *Aspergillus*, and *Colletotrichum* at high frequency. *Diplodia* has been reported as a pine pathogen (*Luchi et al., 2014*), and *Colletotrichium* is a well-known phytopathogen (*da Silva et al., 2020*). *Fusarium* and *Aspergillus* are known plant pathogens (*Syed, Tollamadugu & Lian, 2020*). These variations in taxonomic and fungal group assignments may be attributed, at least in part, to the different types of databases used by each workflow, such as fungal genomes *vs* ITS genetic markers. For example, fewer and sometimes different taxonomic groups were detected when we used a genome-based database in the Kraken2 workflow (manually constructed with fungal taxa; Table S8).

Alpha diversity results varied greatly among workflows, and patterns were not consistent. This can be explained because each workflow represents the results of different taxonomic and functional groups, which do not necessarily follow the same patterns. The only consistent result was the lower diversity in the 11 and 12 sampling points (sites closer to the *Iztaccíhuatl* volcano). Similarly, no significant correlations were found between the

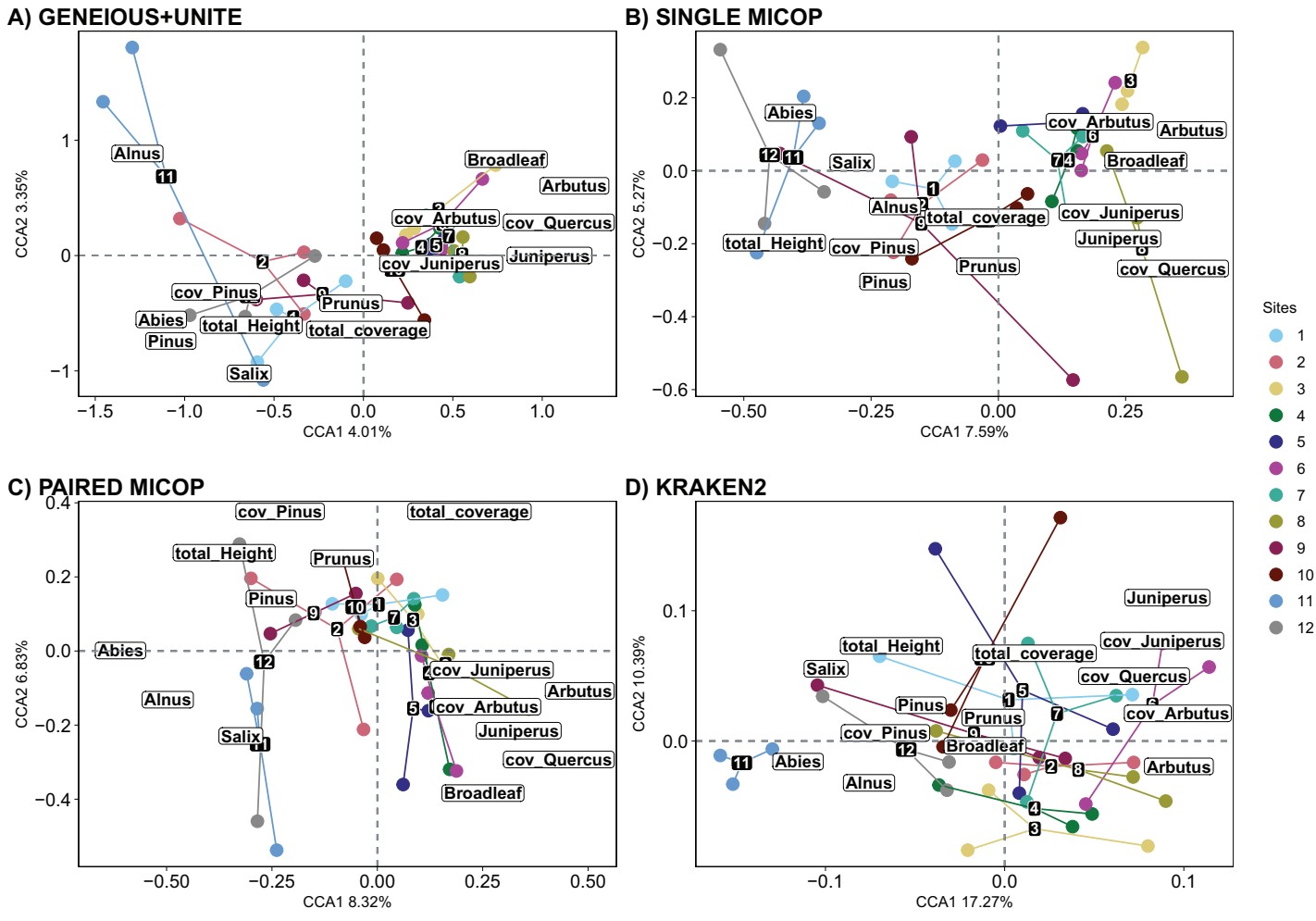

**Figure 8 Canonical correspondence analysis (CCA) of the composition of soil fungal groups and vegetation traits**, *i.e.*, proportion of genera, total coverage and coverage by genus, and total height of the trees, using different workflows: (A) GENEIOUS+UNITE, (B) SINGLE MICOP, (C) PAIRED MICOP and (D) KRAKEN2 (see Fig. 2). Numbers refer to the different sampling sites and their location is given in Fig. 1.

alpha diversity obtained with the different workflows and the factors analyzed in this study (spatial distance, vegetation, and edaphic properties; data not shown).

Since the results of the different workflows detected different functional and taxonomic groups, we used these data to test hypotheses related to different functional groups of fungi. We found similar clustering patterns for the symbiotrophs (Geneious+UNITE), pathogens (Paired MiCoP), and saprotrophs (Single MiCoP), suggesting that environmental, vegetation, and dispersal limitation affected the composition of these fungal guilds to a similar extent. Clustering patterns when considering saprotrophs +pathogen fungi (Kraken2), however, were unclear.

Our study observed that sampling points closer to the National Parks clustered. This clustering can be attributed to the shared environmental and vegetation characteristics among these sites (Figs. S4 and S6). Although previous studies have identified geographical distance as an important predictor of soil fungal community composition globally

(*Tedersoo et al., 2014*), spatial distance or dispersal limitation is not a major factor at the regional scale (*Xing et al., 2021*). Instead, the determinant factors at the regional scale are primarily related to environmental and vegetation/host characteristics (*Peay, Baraloto & Fine, 2013*; *Tedersoo et al., 2020*; *Xing et al., 2021*). However, it is worth noticing that our study revealed a decay in similarity with spatial distance, which can be attributed to the geological relief of the mountains (*Bahram et al., 2013*). Additionally, landscape heterogeneity resulting from land-use changes, such as those affecting the study areas, can decrease gene flow by creating geographic barriers or a matrix of unsuitable habitats that restrict dispersal (*Bisbing et al., 2021*). This suggests that dispersal limitation may still contribute to shaping fungal community composition and distribution, albeit not being the primary factor in our study.

The analysis of environmental characteristics showed that sites 1, 2, 9, 10, 11, and 12, closer to the National Parks, were associated with vital characteristics for forest environments, such as soil SOM, P, and N content. Intermediate sites 3, 4, 5, 6, 7, and 8 were associated with pH, Mg, Ca, Mg, Fe, WHC, Clay, Silt, and Cu. Intermediate sites were located in forest fragments surrounded by agricultural land. This spatial arrangement provides a plausible explanation for the observed patterns. Previous studies have reported a decline in N and SOM content when land use changes, *i.e.*, from forest to agriculture (*Campos et al., 2007*; *Steenwerth et al., 2002*). Additionally, higher values for pH, and Ca and Mg concentrations have been reported in arable soil. This is attributed to the application of fertilizers, pesticides, and chemicals, such as calcium carbonate ($CaCO_3$), contributing to the specific edaphic characteristics of these sites (*Pouyat et al., 2007*; *Steenwerth et al., 2002*).

The pH, and SOM, loam, Ca, and Mg content were identified as common drivers of the fungal groups across all datasets, *i.e.*, symbiotrophs (Geneious+UNITE), saprotrophs (Single MiCoP), pathogens (Paired MiCoP) and saprotrophs+pathogens (Kraken2). For saprotrophs and pathogens, *i.e.*, as determined with Single MiCoP, Paired MiCoP, and Kraken2, total N emerged as an important driver. Although N content has been reported as an important driver shaping EcM taxa (*Yang et al., 2022*), we did not find this effect for symbiotrophs (Geneious+UNITE) in this study. The results obtained from Single MiCoP and Paired MiCoP workflows revealed SOM as the defining characteristic shaping saprotrophs. They specialize in decomposing organic matter, which makes it an important factor in their distribution and abundance.

The significance of pH as a driver of fungal communities on a regional scale agrees with previous reports (*Tedersoo et al., 2020*; *Vasco-Palacios et al., 2020*). pH is a fundamental factor in the assembly and distribution of soil microorganisms across various spatial scales, from local to global (*Aslani et al., 2022*; *Ge et al., 2017*; *Genevieve et al., 2019*; *Glassman, Wang & Bruns, 2017*; *Liu et al., 2018*). It controls the capacity of soils to store or release nutrients as it affects the solubility of P, Fe, Mn, Al, and Ca. Soil pH is closely associated with soil Ca content and tree species' abundance, which affect fungal richness and composition (*Tedersoo et al., 2020*). Other factors, such as SOM or N content, emerge as important drivers of fungal communities, but their significance may vary based on the
location of the studied area. It would be interesting to investigate whether Ca consistently affects fungi across different scales and locations.

The number of vegetation traits that accounted for the taxonomic composition varied across the different datasets. For symbiotrophs (Geneious+UNITE), 12 vegetation traits were found to be significant, while nine were identified for saprotrophs (Single MiCoP), seven for pathogens (Paired MiCoP) and four for saprotrophs+pathogens (Kraken2). These findings align with previous studies indicating that vegetation plays a more prominent role in shaping the community structure of biotrophic fungi, such as symbiotrophs, compared to that of saprotrophic or pathogenic fungi (*Janowski & Leski, 2022*; *Tedersoo et al., 2020*). In contrast, vegetation composition and configuration had a smaller effect on saprotrophs and pathogens. If the pathogenic fungi we identified had remained pathogens of plants instead of animals, then vegetation would have significantly influenced this fungal guild. Comparing the patterns identified in this study concerning saprotrophic fungi with other investigations is challenging, primarily because most studies tend to prioritize the analysis of symbiotrophic fungi, with less focus on saprotrophic fungi.

Notably, sites near both National Parks were characterized by coniferous vegetation, while broadleaf trees dominated intermediate sites. Previous studies at the regional scale have highlighted the importance of the relationship between fungal communities and vegetation in maintaining biological diversity (*Yang et al., 2017*). Our findings support the significance of this association and propose that aboveground plant characteristics could serve as a valuable proxy for regional conservation planning. Therefore, considering vegetation characteristics is crucial when studying the distribution and diversity of fungal communities.

Interestingly, the genera *Russula*, *Sebacina*, *Inocybe*, *Tomentella* and *Cortinarius* are often dominant across the Nearctic, Palearctic and Neotropic zones (*Barroetaveña, Cázares & Rajchenberg, 2007*; *Corrales et al., 2022*; *Erlandson et al., 2016*; *Li et al., 2018*; *Reverchon et al., 2012*; *Tedersoo et al., 2022*). Despite variations in their species assemblage, these fungi display a consistent global presence in diversity, prevalence, and dominance. The high abundance of *Diplodia* in the studied soil is a cause for concern, as it is a pathogen affecting trees and crops (*Luchi et al., 2014*). Our findings suggest that the soil may serve as a reservoir for propagules of this important phytopathogen. *Enterocytozoon* is the most common microsporidia infection. Microsporidia, vertebrate intestinal commensals, are single-celled parasites with a broad host range and can become opportunistic pathogens. While not previously detected in soil metagenomic analyses (*Donovan et al., 2018*), our study reveals its high abundance, which is unexpected given its life cycle. However, it might have originated from vertebrate feces as they were detected as propagules and not in their active form.

The National Parks used in this study are in areas affected by the megalopolis comprising Mexico City and its surrounding areas, whose rapid growth has significantly impacted the natural ecosystems. The extensive fragmentation and disturbance of natural vegetation have led to a potential loss of biodiversity (*Tobón et al., 2016*). It is, therefore, crucial to understand the diversity and distribution of species in this region and identify

areas of high species richness (hotspots) and conservation areas to preserve its biodiversity. Fungal communities play a vital role in forest ecosystems, so it is essential to identify the factors that shape their composition and distribution, considering the potential negative impact of anthropogenic activities on them.

Metabarcoding approaches have been employed to obtain more information from environmental samples. These methods have limitations, such as time-consuming protocols and biases introduced by specific primers used. In contrast, shotgun metagenome sequencing is increasingly used. This method analyzes total eDNA, allowing for studying fungi, bacteria, viruses, and other organisms and their interactions, potential functions, and community structure. However, metagenomics is more costly, bioinformatically complex, and requires deep sequencing to reliably represent less abundant organisms, such as fungi. Several algorithms and programs to analyze metagenomic data have been developed in recent years, including BLAST, USEARCH, DIAMOND, and Kraken (*Alloui et al., 2015*; *Buchfink, Xie & Huson, 2015*; *Camacho et al., 2009*; *Wood, Lu & Langmead, 2019*), which identify the most similar sequence of their database and trace it to a lower common ancestor (LCA). Additionally, new approaches such as MiCoP utilize fast-read mapping to construct a comprehensive reference database of full eukaryotic genomes, maximizing read usage (*LaPierre et al., 2019*). Other tools, such as QIIME2 (*Bolyen et al., 2019*), can be used if reads are first mapped against a reference database with programs like Geneious (*Kearse et al., 2012*). This approach has been used in species of lichens that are not frequently found in reference databases (*Smith et al., 2020*). Still, a significant challenge is the limited availability of pipelines or workflows for studying fungal communities compared to those for viral and bacterial communities (*LaPierre et al., 2019*; *Taş et al., 2021*).

Bioinformatic workflows for taxonomic annotations of metagenomic data have fundamental differences. Some workflows like Kraken2 and MiCoP use fungal genome reference databases. This approach allows to identify almost all fungal metagenomic reads as fungal. Still, a drawback is that the limited number of available fungal genomes means that only a small number of taxa can be detected. On the other hand, other methods use the ITS marker gene database (*e.g.*, UNITE+Geneious), which can identify a larger number of taxa due to the extensive ITS sequence data available. However, it is limited to detecting only ITS regions in the metagenomes. Our results reflect these differences: MiCoP and Kraken2 identified more reads but fewer taxa, whereas UNITE+Geneious identified more taxa but fewer reads. Due to these fundamental differences among the methods, we can assume that results obtained will be different.

## CONCLUSIONS

We used four different bioinformatic workflows to profile fungal taxonomy and found variations in the results depending on the workflow used. Each workflow emphasized different characteristic fungal groups, *i.e.*, symbiotrophic, saprotrophic and pathogenic fungi. Environmental, vegetation and geographical factors affected the spatial distribution of soil fungi on a regional scale. pH, and SOM, Ca, Mg and loam content were common drivers of fungal communities independent of the workflow used. Vegetation traits shaped

the composition of symbiotrophic fungi more than that of saprotrophic and pathogenic fungi. This highlights the importance of considering vegetation traits when studying fungal community diversity and distribution. Clustering patterns of sampling points near National Parks, indicating shared environmental and vegetation traits. Dispersal limitation was found to contribute to fungal community composition, although it was not the primary factor in the study. Overall, this study provides important insights into the challenges and opportunities of studying fungal communities at a regional scale using metagenomic data. However, the workflow should be reported as it might be biased towards specific fungal guilds.

## ACKNOWLEDGEMENTS

To all ejidos, landowners, managers, and authorities who allowed us to conduct this study, Special thanks to ejido San Rafael Ixtapalucan, ejido San José Nanacamilpa, ejido Moxolahuac, ejido Hueyotlipan, Ejido San Gabriel and ejido La Caridad Cuaxonacayo. To the owners and managers of Tandi-Chala Adventure Park and Camping, Piedra Canteada Ecotourism Park and Paraje el Madroño. To the communities of Álvaro Obregón, San Francisco Mitepec, Tizatlán, el Carmen las Carrozas, La Ascensión Huitzcolotepec, San Ambrosio Texantla, San Francisco Tetlanohcan, San Rafael Tepatlaxco, San Esteban Tizatlán, Santa Cruz Tenancingo and San Felipe Hidalgo. To the authorities of the 16 municipalities (belonging to the states of Tlaxcala, Puebla and the State of Mexico): Amaxac de Guerrero, Chiautempan, Españita, Hueyotlipan, Ixtacuixtla de Mariano Matamoros, Ixtapaluca, Nanacamilpa de Mariano Arista, Panotla, San Francisco Tetlanohcan, Tlahuapan, Teolocholco, Tlaxcala, Totolac and Xaltocan. To the authorities of "*Comisión Nacional de Áreas Naturales Protegidas*" (CONANP) for allowing us to work in the National Parks, to Secretaría de Medio Ambiente de Tlaxcala. To the project technicians M. en G. Lenin Ríos Figueroa and M. en C. María Chanel Juárez Ramírez that helped us in the field, with logistics and administrative matters. To Judith Galicia for her support with the administration of the project grant. The authors thank "*Estación Científica La Malinche*" for their logistic support during fieldwork; Ph.D. Ricardo Vazquez Juarez, "*Laboratorio de Genómica y bioinformática*" of CIBNOR and "*Microbioma Lab*" for his help in the bioinformatics analysis and access to computing resources.

### Funding

This research was funded by "Fondo Sectorial de Investigación de la Secretaría de Relaciones Exteriores y el Consejo Nacional de Humanidades Ciencias y Tecnologías" (SRE-CONAHCYT) (Grant CAR 286794). Stephanie E Hereira-Pacheco received grant-aided support from CONAHCYT (grant number: 929602). The funders had no role in study design, data collection and analysis, decision to publish, or preparation of the manuscript.

## Grant Disclosures

The following grant information was disclosed by the authors:

Fondo Sectorial de Investigación de la Secretaría de Relaciones Exteriores y el Consejo Nacional de Humanidades Ciencias y Tecnologías (SRE-CONAHCYT): CAR 286794. CONAHCYT: 929602.

## Competing Interests

The authors declare that they have no competing interests.

## Author Contributions

- Stephanie Hereira-Pacheco performed the experiments, analyzed the data, prepared figures and/or tables, authored or reviewed drafts of the article, and approved the final draft.
- Itzel Arias-Del Razo conceived and designed the experiments, authored or reviewed drafts of the article, funding acquisition and project administration, and approved the final draft.
- Alejandra Miranda-Carrazco analyzed the data, prepared figures and/or tables, authored or reviewed drafts of the article, and approved the final draft.
- Luc Dendooven analyzed the data, authored or reviewed drafts of the article, funding adquisition, and approved the final draft.
- Arturo Estrada-Torres conceived and designed the experiments, authored or reviewed drafts of the article, and approved the final draft.
- Yendi E. Navarro-Noya conceived and designed the experiments, performed the experiments, analyzed the data, authored or reviewed drafts of the article, and approved the final draft.

## DNA Deposition

The following information was supplied regarding the deposition of DNA sequences:

The raw sequences are available at SRA NCBI: PRJNA875082.

## Data Availability

All scripts and metadata for statistical analyses and visualizations are available at Zenodo: Stephanie Elizabeth Hereira Pacheco. (2024). Steph0522/Fungal_Metagenomes_PNIP_PNML: Latest version final (Final). Zenodo. https://doi.org/10.5281/zenodo.13647231.

## Supplemental Information

Supplemental information for this article can be found online at http://dx.doi.org/10.7717/peerj.18323#supplemental-information.

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
