# Peer review of "Metagenomic analysis of fungal assemblages at a regional scale in high-altitude temperate forest soils: alternative methods to determine diversity, composition and environmental drivers"

_PeerJ, doi:10.7717/peerj.18323_

## Round 0.1 · original submission · Major Revisions

The authors are requested to carefully revise the manuscript and answer the questions raised by the reviewers.

Reviewer 1 ·

Basic reporting

See additional comments

Experimental design

See additional comments

Validity of the findings

See additional comments

Additional comments

In this paper, Hereira-Pacheco and colleagues present a metagenomic analysis of soil fungal communities in montane ecosystems of Mexico investigating the spatial and environmental determenets of community composition and putative function. The authors make significant effort to analyze their data via four different bioinformatic pathways, which leads to different conclusions based on the approach taken. This work is interesting and timely, shedding light on the factors structuring these understudied ecosystems. However, the presentation of this research could be improved, and some additional analysis could likely also better answer the questions being asked.

Major comments:

1. This introduction is a bit disorganized and lacks sufficient scholarship in some places. For example, the “main character” of this study, soil fungi, are not introduced until the second paragraph. I think more effort should be made to guide the reader through the knowledge gap you wish to outline. Further, there are some sweeping statements that I feel are inappropriate or incorrect. For example, at L67: “Despite the significance of soil fungi, our understanding of their spatial patterns and the ecological factors that shape their communities remains largely unknown.” This is an overstatement, we have quite a bit of insight into patterns of spatial and ecological inference on microbiomes. The problem is that these are largely context specific, with low generalizability across ecosystems and large spatial scales. Emphasizing this point would significantly strengthen your manuscript. Additionally, at L 80: "While most studies focus on global fungal distribution (Tedersoo et al., 2010; 2014), a limited number are centered on local and regional scales (Bahram et al., 2015; Tedersoo et al., 2020).” It is actually the opposite. Far more studies focus on local and regional distribution of fungi than global, for example:

Glassman, S. I., Wang, I. J., & Bruns, T. D. (2017). Environmental filtering by pH and soil nutrients drives community assembly in fungi at fine spatial scales. Molecular ecology, 26(24), 6960-6973.

Zhao, J., Gao, Q., Zhou, J., Wang, M., Liang, Y., Sun, B., ... & Yang, Y. (2019). The scale dependence of fungal community distribution in paddy soil driven by stochastic and deterministic processes. Fungal Ecology, 42, 100856.

I found these after a very short time of googling “Local distribution soil fungi”. Please do better scholarship, and I would encourage you to look outside the Tedersoo group. The authors tend to rely heavily on some core citation (i.e., Tedersoo et al 2020), which may be significantly limiting your view of the overall field.

2. The two main goal of this study (what’s affecting fungi vs which bioinformatic approach is best) are a bit at odds with each other in the text, making it more difficult to graft the finding overall. Often, bioinformatic decisions are made a priori based on the type of sequencing and the conventions of others in the field. More care should be taken to weave these two components together to tell a more compelling story about the ecosystem its self, rather than presenting a list of which method did what.
3. For disentangling spatial versus environmental drivers, I would highly encourage you to look into general dissimilarity modeling and Mantel tests. I think these approaches would be more effective at elucidating the importance of various factors at increasing scales.

Minor comments:

98: Why are you referencing the Baltic region specifically in this section? Could you articulate how this connects to your system a little better?

113: Could you explain a bit more about these biases? How do you know there was a bias introduced, or why might the biases have been geared towards specific groups based on the bioinformatic approach?

164: What method? Just a bit more detailed description here

176: Need more detail about library preparation.

248: This should likely be a mantel test.

437: Tedersoo et al 2020 is cited 5 times in this paragraph, please look more broadly for literature I this field.

Figure 5: Could you show the variation explained by each of the axis on these ordinations? Same for other ordinations below.

Figure 6: A partial mantel test using the ecological distance of plants and fungi, with the Euclidian distance of space and soil traits would be a more effective way to analyze this data.

Reviewer 2 ·

Basic reporting

The English language, structure of the manuscript, and figures are adequate. Raw data are supplied. The background information in the Introduction section is relevant but more context in some areas is needed (see General Comments).

Experimental design

There needs to be additional context and justification for why these specific methods were chosen for the methods comparison (see General Comments). Other than that, I think the methods are adequate and described in sufficient detail.

Validity of the findings

Underlying data and analysis code are provided. Findings appear to be adequately supported, though the meaningfulness of the comparisons between the different methods needs more discussion (see General Comments).

Additional comments

General Comments

In this manuscript, the authors used four different metagenomic workflows to characterize soil fungal biodiversity in the Trans-Mexican Volcanic Belt. Though the different bioinformatic workflows produced very different results, they generally found that vegetation characteristics and soil properties such as pH, Ca, Mg, and silt content to be the dominant drivers of fungal biodiversity in the region. This study will be of interest to investigators interested in using soil metagenomes to characterize fungal communities as well as investigators interested in this particular biogeographic region. My comments on the manuscript are found below.

Since this paper is primarily a methods comparison among different bioinformatics approaches, my major comments relate to the methods. I think there needs to be much more context for and justification of the specific methods chosen. My comments are below.

• The ‘QIIME2’ method needs to be renamed. QIIME2 is not a specific bioinformatics workflow, rather it is an environment in which many different analysis options are available. For example, the Kraken2 method the authors used is also provided in QIIME2. The specific method they are referring to here is mapping to UNITE with Geneious. The name should reflect that.
• There should be more information provided on why these specific methods were compared. There are many different bioinformatics workflows that assign taxonomy to reads. Why were these specific methods chosen for comparison?
• It is unusual to use metagenomics to characterize fungal biodiversity. Usually this would be done using amplicon sequencing, e.g., of one of the ITS regions. The reason for this is that only a very small proportion of metagenomic reads from soil will be fungal in origin (most will be prokaryotic). This is directly evident in the results shown in Table 1. It appears that very few reads per sample were identified as fungal. I think the authors should include a column in the table showing the percentage of the original reads that were used in the fungal analyses. Along this same line of reasoning, I think the authors should include species accumulation curves (rarefaction curves) for each of their methods to determine if fungal communities were adequately sampled. My guess is that none of the methods will produce saturated rarefaction curves. I think this is important information to include.
• I think there needs to be a lot more context on the fundamental conceptual differences between the different bioinformatic workflows. Some of the workflows (Kraken2, MiCop) use fungal genomes reference databases. The advantage of this is that any metagenomic read could be identified as fungal but the disadvantage is that few fungal genomes are available, so few taxa can be found. In contrast, the authors’ QIIME2 (really UNITE) method uses only the UNITE ITS marker gene database. The advantage to this is that many more taxa can be found (since there are many taxa with ITS sequences) but the disadvantage is that only ITS genes in the metagenomes can be found. These differences are directly reflected in the authors’ results – MiCoP and Kraken2 identified more reads but few taxa and vice versa for QIIME2 (UNITE). Given these fundamental differences among the methods, one would not expect to get comparable results, so the meaningfulness of these comparisons is not clear to me. I think more discussion of this is necessary.

Minor Comments

• Line 95: I disagree with this. There are many studies showing that AMF respond to land use changes, e.g., forest management practices
• Line 112: The methods certainly produce different results, but I do not think it is correct to call them all “biased.” This would imply that the true community composition was known and that the methods all differed from the known composition. I think it is more accurate to simply call them ‘different’ rather than ‘biased’
• Line 143: This should be ‘O horizon’
• Line 154: I assume here that the soil was first dried before weighing to determine WHC. I think that would be the only way to determine the water content. Please clarify.
• Line 162: Is this total soil N or total extractable N?
• Line 164: This method of determining SOM should be briefly described here.
• Line 247: In my opinion, distance-decay is not a perfect indicator of dispersal limitation. It is possible to directly quantify the contribution of dispersal limitation using phylogenetic null models, which might be a feasible approach for this dataset.
• Line 261: In my opinion, hierarchical partitioning is a simpler and more effective way to obtain this information compared with the CCA/RDA method described here
• Line 388: They are not just different databases, but fundamentally different types of databases (i.e., fungal genomes vs. ITS sequences, see general comments)
• Like 491: The authors also need to discuss the disadvantages of using shotgun metagenomes for this purpose, i.e., the very small proportion of reads that will be fungal.

·

Basic reporting

This manuscript is well-written. Sufficient background and literature references are provided, and it is self-contained with all relevant results. English is clear and professional throughout. Sequence data is provided via NCBI’s Short Read Archive, and code and additional data is available via GitHub.

Experimental design

In this manuscript, the authors assess the distribution and diversity of fungal communities in the Trans -Mexican Volcanic Belt (TMVB) region via metagenomics. They hypothesized that different factors would drive the composition and distribution of different fungal guilds. This hypothesis was supported by their results, which found that vegetation type was an important driver of ectomycorrhizal fungi, but not other types of fungi. The authors compared four different bioinformatics pipelines for classification of metagenomic reads; Kraken2, QIIME2, and MiCoP (both single- and paired-end methods). While each of these pipelines produced different results, all results were in line with previous findings of soil fungal communities and did not appear erroneous. Overall, this research is well-planned and well-performed. All methods used are standard in the field and described in sufficient detail for replication.

To me, the most valuable contribution of this paper is the direct comparison of four different classification methods. I was surprised to see how different the results from each method were! I agree with the authors’ assessment that none of the methods are “wrong” or “right” – they all just have different biases, and it is important to understand these biases when selecting a method. The one weakness I see in this comparison is that each method uses a different reference database, and this alone could bias the results of classification. The most direct comparison would be to use the same database for all of the classification methods; however, changing reference databases for QIIME2 and MiCoP is not trivial, and this analysis would likely be beyond the scope of this work. The authors describe this in L388 and mention that they tried a second, smaller database for Kraken2 with different results and data not shown – I would be interested in seeing a very brief summary of the differences found in supplemental data. I will certainly be keeping this paper in mind when planning my own metagenomic studies!

Validity of the findings

There is sufficient information included to replicate these methods and clear benefit to the literature. I appreciate the use of GitHub to make all code available. Conclusions are well-stated and appropriate for the results obtained.

Additional comments

Abstract: include that each method found different fungal groups – to me, this is a major finding of your paper!
L111: “established” isn’t the right word here – to me, that would mean that you created these methods or were the first to publish them. “used” would be more appropriate.
L149: missing “was” – “while the rest WAS air-dried and characterized”
L242: citation for R package ape missing? (Thanks for including citations for R packages throughout the methods. The developers appreciate it!)
Fig 3 legend: capitalize names of bioinformatics software
Fig 5 legend: “principle” instead of “principal”
Fig 3: There’s a lot of information in this figure, and it’s hard to read the species names and connect them to the right color for each data type. Consider moving the guild and trophic status to a supplemental figure instead. Also, I assume the tree on the left-hand side represents similar distribution patterns and not phylogeny? If so, consider removing as it does not add any more information to the heatmap.

·

Basic reporting

The manuscript by Hereira-Pacheco et al. offers valuable insights into the diversity and distribution of fungal communities in neotropical mountainous forest soils. I have a few minor comments, and addressing these suggestions will enhance the manuscript’s contribution to the field of fungal community ecology in this ecosystem.
Line 39-42: Why does vegetation traits had differential impacts on symbiotrophic versus saprotrophic and pathogenic fungi. Any ecological implications?
Line 43-45): The authors should provide a brief interpretation of the spatial clustering near volcanic regions and the implications of distance decay in taxonomic similarity
Lines 50-51: The authors should explore more recent literature review on the relevance of mountain forests in the neotropics. For instance, mentioning their global importance as biodiversity hotspots or their vulnerability to climate change could enhance this section.
Lines 53-56: Any specific ecological threats faced by TMVB due to its unique biogeography?
Lines 67-79: The discussion on gaps in understanding fungal community spatial patterns and ecological drivers is relevant and well-supported. I think the authors should consider expanding on why understanding these patterns is crucial for ecosystem management or conservation strategies in TMVB.
Lines 126-137: Why did you select LMNP and IPNP specifically within TMVB. Any specific ecological characteristics?
Lines 138-146: I suggest unit consistency (e.g., km² vs. km2) throughout the manuscript. Again, the authors need to tell us how the removal of horizon O and sampling of 0-10 cm topsoil aligns with standard practices in soil ecology and how this might impact fungal community analysis.
Lines 179-208: software version?
Lines 269-278: Please provide a succinct summary of key findings related to richness and diversity across different sites. Its interesting but the authors did not really elaborate on why Kraken2 showed low variation compared to other workflows.
Lines 304-316: Please consider discussing potential ecological implications of clustering patterns observed. It seems the authors could link dissimilarity patterns more explicitly to environmental or geographical factors influencing fungal communities?
Lines 325-347: I will like the authors to compare their findings with similar studies in different geographic regions.

Experimental design

The manuscript by Hereira-Pacheco et al. offers valuable insights into the diversity and distribution of fungal communities in neotropical mountainous forest soils. I have a few minor comments, and addressing these suggestions will enhance the manuscript’s contribution to the field of fungal community ecology in this ecosystem.
Line 39-42: Why does vegetation traits had differential impacts on symbiotrophic versus saprotrophic and pathogenic fungi. Any ecological implications?
Line 43-45): The authors should provide a brief interpretation of the spatial clustering near volcanic regions and the implications of distance decay in taxonomic similarity
Lines 50-51: The authors should explore more recent literature review on the relevance of mountain forests in the neotropics. For instance, mentioning their global importance as biodiversity hotspots or their vulnerability to climate change could enhance this section.
Lines 53-56: Any specific ecological threats faced by TMVB due to its unique biogeography?
Lines 67-79: The discussion on gaps in understanding fungal community spatial patterns and ecological drivers is relevant and well-supported. I think the authors should consider expanding on why understanding these patterns is crucial for ecosystem management or conservation strategies in TMVB.
Lines 126-137: Why did you select LMNP and IPNP specifically within TMVB. Any specific ecological characteristics?
Lines 138-146: I suggest unit consistency (e.g., km² vs. km2) throughout the manuscript. Again, the authors need to tell us how the removal of horizon O and sampling of 0-10 cm topsoil aligns with standard practices in soil ecology and how this might impact fungal community analysis.
Lines 179-208: software version?
Lines 269-278: Please provide a succinct summary of key findings related to richness and diversity across different sites. Its interesting but the authors did not really elaborate on why Kraken2 showed low variation compared to other workflows.
Lines 304-316: Please consider discussing potential ecological implications of clustering patterns observed. It seems the authors could link dissimilarity patterns more explicitly to environmental or geographical factors influencing fungal communities?
Lines 325-347: I will like the authors to compare their findings with similar studies in different geographic regions.

Validity of the findings

The manuscript by Hereira-Pacheco et al. offers valuable insights into the diversity and distribution of fungal communities in neotropical mountainous forest soils. I have a few minor comments, and addressing these suggestions will enhance the manuscript’s contribution to the field of fungal community ecology in this ecosystem.
Line 39-42: Why does vegetation traits had differential impacts on symbiotrophic versus saprotrophic and pathogenic fungi. Any ecological implications?
Line 43-45): The authors should provide a brief interpretation of the spatial clustering near volcanic regions and the implications of distance decay in taxonomic similarity
Lines 50-51: The authors should explore more recent literature review on the relevance of mountain forests in the neotropics. For instance, mentioning their global importance as biodiversity hotspots or their vulnerability to climate change could enhance this section.
Lines 53-56: Any specific ecological threats faced by TMVB due to its unique biogeography?
Lines 67-79: The discussion on gaps in understanding fungal community spatial patterns and ecological drivers is relevant and well-supported. I think the authors should consider expanding on why understanding these patterns is crucial for ecosystem management or conservation strategies in TMVB.
Lines 126-137: Why did you select LMNP and IPNP specifically within TMVB. Any specific ecological characteristics?
Lines 138-146: I suggest unit consistency (e.g., km² vs. km2) throughout the manuscript. Again, the authors need to tell us how the removal of horizon O and sampling of 0-10 cm topsoil aligns with standard practices in soil ecology and how this might impact fungal community analysis.
Lines 179-208: software version?
Lines 269-278: Please provide a succinct summary of key findings related to richness and diversity across different sites. Its interesting but the authors did not really elaborate on why Kraken2 showed low variation compared to other workflows.
Lines 304-316: Please consider discussing potential ecological implications of clustering patterns observed. It seems the authors could link dissimilarity patterns more explicitly to environmental or geographical factors influencing fungal communities?
Lines 325-347: I will like the authors to compare their findings with similar studies in different geographic regions.

Reviewer 5 ·

Basic reporting

This manuscript investigates fungal communities in high-altitude temperate forests of Mexico using metagenomic techniques. The authors explore the diversity and distribution of fungi, comparing the outcomes of four different bioinformatic workflows (QIIME2, MiCoP, and Kraken2) and analyzing the influence of environmental factors like soil chemistry and vegetation on fungal community composition. The study offers insights into the complexity of fungal ecology and suggests methods to enhance understanding of these communities in natural settings. However, there are some issues for improvement in the manuscript:

Major comments:
1. The figures need to be improved and provide the reproducible script in GitHub. Such as ImageGP (https://doi.org/10.1002/imt2.5) can generate high quality figures and with reproducible scripts.
2. The structure of the results appears not in concise. The authors should have presented their findings in around 3 mainly sections will be better.
3. For statistical analysis, EasyAmplicon have better reproducible steps and scripts is recommended. A network compare with each group is also recommended, such as ggClusterNet, iNAP are good tools for network analysis and visualization.
Minors comments:
4. The manuscript could benefit from a brief introduction of the four bioinformatic workflows used (QIIME2, MiCoP, and Kraken2), highlighting their strengths and limitations.
5. Line 247-249, Pearson’s correlation tests were used to evaluate the association between community similarity and geographic distance. Given the nature of ecological data, please check if Spearman’s rank correlation might be more suitable for your data, due to its robustness against non-linear relationships and non-normal distributions.
6. Line 176-177, specify the primers used for library preparation
7. The figures are informative but could be improved for better readability. For instance, some graphs have overlapping texts that are hard to interpret, e.g. Fig 5,7 and 8.

Experimental design

NA

Validity of the findings

NA

---

## Round 0.2 · accepted · Accept

After revisions, three reviewers agreed to publish the manuscript. I also reviewed the manuscript and found no obvious risks to publication. Therefore, I also approved the publication of this manuscript.

Reviewer 1 ·

Basic reporting

Clear reporting of data and contextualization within the field.

Experimental design

Nice questions, experimental design, and analysis

Validity of the findings

Important context for future bioinformatic decision making.

Additional comments

Good work with these revisions. Some very minor comments:

26: To determine what about fungal communities?

147: Was the sieve sterilized between samples/ sites?

149: How were samples stored during transportation from the field to the laboratory and how long did it take between collection and storage at -20oC?

·

Basic reporting

No comment

Experimental design

No comment

Validity of the findings

No comment

Additional comments

The authors have sufficiently addressed my comments, including providing additional results as a supplemental table. Edits in the text improve understanding and clarity. Figures have been modified as requested to improve readability.

Reviewer 5 ·

Basic reporting

No comment.

Experimental design

No comment.

Validity of the findings

No comment.

Additional comments

No comment.